# Nanofibrillar Green Composites of Polylactide/Polyhydroxyalkanoate Produced in Situ Due to Shear Induced Crystallization

**DOI:** 10.3390/polym11111811

**Published:** 2019-11-04

**Authors:** Iurii Vozniak, Ramin Hosseinnezhad, Jerzy Morawiec, Andrzej Galeski

**Affiliations:** Centre of Molecular and Macromolecular Studies Polish Academy of Sciences, 90-363 Lodz, Poland; ramin.h@cbmm.lodz.pl (R.H.); jemor@cbmm.lodz.pl (J.M.); andgal@cbmm.lodz.pl (A.G.)

**Keywords:** green polymer-polymer nanocomposites, shear induced crystallization, in situ fibrillation, crazing-shear banding transition, mechanical properties

## Abstract

This study addresses the new concept of in situ inducing fibrillar morphology (micro or nanofibrils) of a minority component based on the simultaneous occurrence of orientation and shear induced crystallization of polymer fibers directly at the stage of extrusion in a single step. This possibility is demonstrated by using two entirely bio-sourced polymers: polylactide (PLA) and polyhydroxyalkanoate (PHA) as components. The shear induced crystallization allowed crystallization of PHA nanofibers immediately after applying high shear rate and elongation strain, avoiding subsequent cooling to initiate crystallization. Shearing of PHA increased non-isothermal crystallization temperature by 50 °C and decreased the temperature range in which the transition from a molten state to a crystallized one occurs by 17 °C. SEM observations demonstrate the successful transformation of the dispersed PHA phase into nanofibrils with diameters of nearly 200 nm. The transition from the droplets of PHA to fibers causes the brittle-to-ductile transition of the PLA matrix at a low concentration of PHA and contributes to the simultaneous increase of its rigidity and strength.

## 1. Introduction

Recently, there has been a lot of attention on the design of so-called “ecocomposites” or “green composites” that are alternative to synthetic polymer composites. As the performance of green polymer composites depends on the physical, melt rheological characteristics of the components, their morphology, and the interfacial interaction strength, the progress is related both to the search for new pairs of matrices and fillers and to development of new methods of production of polymer composites.

The most widely applied methods of production of green composites are those based on the blending of polymers and ready-made inorganic or natural-organic fibers or particles [1,2,3,4] as well as more complex multi-stage methods addressing in situ formation of varied morphology of the filler: discrete or entangled fibers, shish-kebab structures (when the crystallization of the matrix polymer occurs on the surface of the formed fibrils), etc. [5,6,7,8]. In the last cases, as a rule, a biopolymer immiscible with the biopolymer matrix is used as the minor phase. The general approach to the formation of polymer fibers in situ within another polymer matrix is the preparation of a polymeric blend at the first stage when one of the polymers is dispersed in the form of droplets (micro- or nano) that are later transformed into fibers under the action of forces. As a rule, hot/cold drawing, rolling or electrospinning is applied here. The resultant composite is characterized by the orientation of both components, so an isotropic composite is formed at the second stage by melting and relaxation of the matrix [5].

It should be noted that three key requirements should be met when selecting polymeric pairs for the production of green composites in situ. First, the constituent polymers (both the majority and minority polymer phases) must have a high draw ratio at the processing temperature to form fibrils. Second, both of the constituent polymers must be processable at certain temperature without the onset of degradation in either polymer; and third, the temperature range between the melting/glass transition temperature of the minority and majority polymer phases must be by at least 40 °C to ensure fibril retention during matrix solidification.

Known examples of such green composites which were successfully prepared according to this method include poly(lactic acid) (PLA)/polybutylene succinate (PBS) [9,10,11], thermoplastic polyurethane elastomers/PLA [12], poly(lactic acid)/poly(e-caprolactone) [13,14,15], thermoplastic starch/PLA [16], poly(l-lactic acid)/poly(glycolic acid) [17], poly(l-lactide)/poly(amide) [18,19], poly((l-lactide)-*co-*(e-caprolactone))elastomer/poly(glycolic acid) [20] pairs.

An alternative in situ method to produce fibrillar green composites is micro-injection moulding which is carried out at extremely high shear rates of the order of 10^5^ s^−1^ [13]. This approach is very promising because the spectrum of polymer couples can be extended, but it is hardly realized in the course of conventional injection molding.

Another extrusion technique used for the production of polymer–polymer composites is the dynamic quenching process [21,22]. In the setup for extrusion, the heating zone in the barrel adjacent to the feeding zone, is set at a high temperature to melt both polymers, whereas near the die, the temperature is set to keep one polymer molten while the other is ready to solidify. Then a component with a higher melting (or glass transition) temperature is solidified on the extrusion path, which leads to fibrous dispersion under a flow field. However, only isotactic polypropylene/high density polyethylene composites were obtained in this way.

Recently, another approach to producing green polymer composites was proposed [11]. It is based on the simultaneous realization of orientation and crystallization of polymer fibers directly at the stage of extrusion in a single step by setting high shear strain and shear rate, which force the crystallization of a polymer at a higher temperature without a succeeding cooling. This approach has been used only for the production of green PLA/PBS composite [11]. Significantly increased ductility and slightly lowered rigidity were provided by the generation of PBS nanofibers as well as the formation of shear bands in PLA during deformation. 

Here we show new PLA-based blends, namely rigid PLA with ductile polyhydroxyalkanoate (PHA), which may form an excellent natural pair of complementary materials, and the possibility of their transformation to a polymer–polymer composite by morphology transformation of the minor polymer phase from droplets to fibers. A specific feature of in situ generated composites lays in the fact that the crystallization of the minor polymer phase (i.e., PHA) is forced by high shear rate and elongation strain and its crystals are crystallized at a temperature higher than the softening temperature of the PLA matrix. A slit die extrusion process is proposed for increasing the residence time in the extruder and facilitating the formation of fibril network at a low concentration of PHA. The role of a formed PHA fiber network in strengthening and toughening the PLA matrix at the same time is discussed by using in situ tensile testing inside the SEM sample chamber.

## 2. Experimental Section

### 2.1. Materials

Commercial grade PLA 4060D, supplied by NatureWorks LLC, with a density of 1.24 g/cm^−3^, *M*_w_ of 120,000 g/mol^−1^, and 18 mol% of d-Lactide content was used as the matrix. Polyhydroxyalkanoate (PHA) with the trade name Mirel P5001 was chosen in order to reinforce the PLA. It has a melting point at 170 °C and has a density of 1.30 g/cm^3^.

### 2.2. Sample Preparation 

Following the procedure reported previously [11], 3 wt.% of PHA was melt blended with PLA (both dried for 8 h at 60 °C). Temperature zones of a twin-screw extruder were set increasingly from 160 to 190 °C, while this temperature gradient descended for a single-screw extruder from 175 °C (feed section) to 135 °C (slit die).

### 2.3. Mechanical and Thermal Properties

Tensile properties of composite, blend, and neat PLA were measured in an Instron-5582 (Universal Testing Machine, High Wycombe, UK) at a strain rate of 5%/min according to ISO 527-2. Strain measurement was performed by the use of a clip-on extensometer. Seven specimens were tested for each sample at room temperature. Rectangular samples with dimensions of 24 × 10 × 0.75 mm^3^ were tested for dynamic mechanical thermal analysis (DMTA) using DMA Q800 (TA Instruments, New Castle, DE, USA) at a heating rate of 2 °C/min. Melting and crystallization kinetics were probed with DSC 2920 (TA Instruments, New Castle, DE, USA) under nitrogen purging condition (20 ml/min). The measurements were carried out by heating and subsequent cooling of samples with a rate of 10 °C/min.

### 2.4. Scanning Electronic Microscopy (SEM)

The morphology of samples, cryogenically fractured along the extrusion direction and coated with gold, was investigated with a JEOL JSM-5500 LV scanning electron microscope (Tokyo, Japan). Further observation was done on the samples which were ultramicrotomed with a diamond knife (Diatome, Switzerland) at −130 °C. Gatan MT200 (Microtest Tensile Stage, Suffolk, UK), connected to the microscope, facilitated in situ observations of the tensile test according to ASTM D638. Specimens were coated with carbon and deformed with a rate of 0.2 mm/min following the procedure described previously [11].

### 2.5. Rheo-Optical Measurements: Shear-Induced Crystallization Test

Shear-induced crystallization of the PHA was scrutinized in the Linkam CSS450 (Optical Shearing System, Surrey, UK) as per [11]. Films of PHA were positioned between two heated glass stages undergoing subsequent treatment under a nitrogen flow: (1) heating to 60 °C above the melting point at (30 °C /min), (2) holding for 5 min to eliminate possible orientation of chains, (3) cooling down with simultaneous shearing at rates 100–1000 s^−1^.

### 2.6. Rheological Measurements

The rheological behavior of materials was examined using a strain-controlled rotational rheometer (ARES LS2, TA Instruments). Uniaxial extension tests of molten samples were performed using extensional viscosity fixture (EVF, TA Instruments) attached to the ARES rheometer (New Castle, DE, USA). The 18 × 10 × 0.7 mm^3^ rectangular specimens were prepared by hot compression molding at 200 °C in the standard mold, provided by TA Instruments with the EVF, or were cut out from extruded tapes. The specimen was uniaxially extended at 100 °C with a constant Hencky strain rate, έ, of 1.0, 2.0, or 4.0 s^−1^. The tensile stress growth coefficient, $\eta_{E}^{+}$ (t, έ), as a function of time, t, at a given Hencky strain rate was measured. The tests for each material were repeated at least eight times, and the results were averaged. Steady-state shear measurements were executed to investigate the viscosity–shear rate relationship. Samples were tested using cone-plate geometry with a cone angle of 0.1 rad and a gap distance of 0.046 mm. Disk-shaped samples were cut out from 1 mm thick hot-pressed films. The shear tests were performed at temperatures 135, 155, and 180 °C and shear rates, γ, ranging from 1 to 300 s^−1^.

## 3. Results and Discussion

Figure 1 shows the effect of the shear rate γ on the conversion degree of PHA. It is seen that as the shear rate increases, the crystallization temperature, *T*_c_, shifts towards higher temperature and the transition from a molten state to a crystallized one proceeds in a narrower temperature range, Δ*T*. In particular, an increase in γ up to 300 s^−1^ results in a shift of *T*_c_ toward higher temperatures up to 137 °C and a decrease in Δ*T* from 38 to 24 °C. The maximum effect is achieved at γ = 850 s^−1^. In this case, *T*_c_ is 156 °C and Δ*T* = 17 °C. The shift in *T*_c_ was smaller for the PHA sheared at constant temperature and subsequently cooled down (Figure 2). The observed effect is due to the fact that the implementation of simultaneous shear and cooling leads to a significant reduction in the time gap between cessation of shearing and the onset of crystallization. The shorter this time gap, the less possibility for the crystal nucleus to re-melt at temperatures above non-isothermal crystallization temperature and for polymer chains to recoil and relax. An increase in the shearing rate promotes an increase in the concentration of nuclei and accelerates the crystallization process. In that way, in situ generated PHA fibers would crystallize directly under a high shear rate without the necessity for cooling below the temperature of their non-isothermal crystallization. The relaxation process minimized in this way would cause the retention of the achieved elongation degree of droplet inclusions into fibers.

Figure 3 shows the morphologies of the cryogenically fractured surfaces of the PLA/PHA blend and in situ generated PLA/PHA composites. The PLA/PHA blend presents typical matrix–droplet morphology. PHA spherical particles and dark holes left by them during fracture are observed on the surface of the blend (Figure 3a). The surface of these particles is smooth with clear borders, suggesting poor compatibility and weak interfacial adhesion between the PLA and PHA phases. Meanwhile, the low concentration of PHA (3 wt.%) makes their dispersion in PLA matrix well-uniform and causes the formation of nano-sized spherical particles of PHA. Most PHA dispersed phases are in the range of 50–450 nm and the averaged size is about 275 nm. Intensive blending (screw speed was 120 rpm) also contributes to the deformation and fracture of the PHA phase and minimizes agglomeration of the shear–split PHA phase.

Compared to the PLA/PHA blend, SEM micrographs of the in situ generated PLA/PHA composite reveal the presence of fibril-matrix morphology (Figure 3b). Almost all the PHA dispersed phase particles are elongated into fibrils due to the presence of a shear force field within the area between the extruder screw and the extruder walls. Detailed mixing mechanisms of immiscible polymers are provided elsewhere [23]. It is already well established that when dispersed minor polymer droplets, immersed in the immiscible matrix, are subjected to high shear rates, they tend to deform and extend into long thin threads. As a result, particle radius decreases while the capillary instabilities at the interface increase. When the capillary number approaches to the critical value, the elongated nanofibrils tend to disintegrate and break up. The most important mechanisms, depending on the viscosity ratio and the degree of elongation, could be the growth of uniform Rayleigh disturbances in the middle part of the thread, end-pinching at the ends, and necking. However, rapid solidification caused by shear induced crystallization prohibits these time-dependent transformations. Thus, continuously elongated in the melt, PHA nanofibrils are stabilized and fixed by crystallization before they may undergo any retraction or disintegration.

This is reasonable because the viscosity ratio between the PLA/PHA blend and neat PLA at low temperatures is, according to [24], in the optimum range, which is (0.1:10), and remains within this range despite the shear thinning which occurs at high shear rates, that is the hydrodynamic force overcomes the cohesive strength of the dispersed PHA droplets (shown in the Appendix A). It is also well known that fibrous droplets could eventually take the form of spherical ones or break up into small particles by interfacial tension, but rapid solidification prohibits these transformations and results in fibrous dispersion.

The final nanofibers are 70–130 nm in diameter and form the physical network structure. The formation of the physical network of PHA in the PLA matrix is possible during compounding and not during extensional flow in the slit die zone. Apparently, the slit capillary serves as a dumping for the flow and to increase the residence time in the extruder and facilitate the formation of fibril entanglements. The ratio of the diameters of the fibrils to the droplets shows that the aspect ratio of PHA fibrils is at least 100. Since the exact length of the fibers is unclear, it can be assumed that the actual aspect ratios of PHA fibrils can be even larger because of weak interfacial adhesion between the PLA and PHA phases and a uniform distribution of PHA nanoparticles, which increase the probability of them coalescing during shearing, and thereby facilitate formation of very long fibrils. The dispersed PHA fibrils show tight contact with the PLA matrix, leaving no distinct voids between the elongated fibers and the matrix. This fact indicates a good adhesion between the fibrils and the matrix.

The SEM micrograph analysis of the PLA/PHA composite in longitudinal and transverse sections shows the uniform distribution of the microfibrils, and nearly the same number of revealed fibrils in both sections indicates the possibility of forming a physical network structure (Figure 4).

According to [25], 2.5 wt.% of the minor phase corresponds to the gelation point at which physical network structure forms. To verify this assumption, the viscoelastic behavior of the PLA/PHA blend and in situ generated PLA/PHA composites were examined. Figure 5 presents a time-dependence of the tensile stress growth coefficient, $\eta_{E}^{+}$ (t, έ), for molten PLA, PLA/PHA blend and composite recorded during uniaxial tension at various Hencky strain rates, έ. For neat PLA and the PLA/PHA blend, $\eta_{E}^{+}$ (t, έ) gradually increases with time, t, but it is nearly independent of έ. The monotonic stress growth reflects a predominantly plastic flow. The tensile stress growth coefficient, $\eta_{E}^{+}$ (t, έ), then reaches the end of plateau when the chains become highly extended. At and beyond this plateau, chains could start to slide past one another at the entanglement points causing yielding of the entanglement network and the appearance of a strain softening effect. As the number of load-bearing entanglements decreases, the tensile stress growth coefficient, $\eta_{E}^{+}$ (t, έ), is reduced as shown in Figure 5. 

In contrast to neat PLA and the PLA/PHA blend, exhibiting no strain hardening itself, the in situ generated PLA/PHA composite containing a PHA fiber network structure demonstrates a significant increase of $\eta_{E}^{+}$ (t, έ) above the linear viscoelastic region during uniaxial extensional deformation. This is because the PHA fiber network, containing physical entanglements between the fibers, is deformed during uniaxial extension causing such an upward deviation of $\eta_{E}^{+}$ (t, έ) for the PLA/PHA composite, known as strain hardening.

An important condition of effective conversion of the droplets of the minor polymer into fibers is the immiscibility of polymer constituents. According to the DSC analysis of the PLA/PHA extruded nanocomposite, melting of 3 wt.% of PHA is evidenced by a small endothermic peak at around 170 °C (shown in the Appendix A). The glass transition of the PLA matrix is seen at around 60 °C with an enthalpy relaxation signal caused by PLA orientation. The separate melting peak of PHA crystals proves that PHA is not miscible with PLA.

A similar conclusion about the immiscibility of PLA and PHA can be drawn based on E^//^ data from DMTA measurements of PLA/PHA extruded tapes (shown in the Appendix A). The E^//^ peak at the PLA glass transition is not shifted for the PLA/PHA nanocomposite and the glass transition of PHA is not discernible at –17 °C because of the low concentration of PHA (3 wt.%).

The morphology of the dispersed phase plays an important role in achieving improved mechanical properties of polymer blends or composites. In the case of the PLA/PHA blend, the spherical PHA particles with a diameter ranging between 50 and 450 nm were found to be distributed in the PLA matrix, which indicates a completely immiscible blend. Additionally, the PHA particles exhibited a uniform dispersion in the PLA matrix. The morphology of immiscible phases is also observed in the case of the PLA/PHA composite, with fibers with a diameter of 70–130 nm and an aspect ratio of not less than 100. As a result, for the PLA/PHA blend, the Young modulus, stress at break, and elongation at break exceeded the related parameters of neat PLA by 5%, 28%, and 6% (2.14 GPa, 55.0 MPa, and 7.4%), respectively. In the case of the in situ generated PLA/PHA composite chatacterized by fibrillar morphology, the Young modulus, stress at break, and elongation at break exceeded the related parameters of neat PLA by 15%, 30%, and 400% (2.35 GPa, 56.0 MPa, and 28.1%), respectively. This could be explained by the presence of PHA nanoparticles in the PLA matrix, which restrict the mobility of PLA chains and thus improve the stiffness of the PLA matrix. Moreover, the absence of signs of aggregation of the PHA nanoparticles in the PLA matrix i.e., the homogeneous dispersion of the nanoparticles, leads to a more regular and stronger structure of the blend, thereby improving the interfacial integrity of PLA and PHA to better transfer the stress under external force [26]. However, the much larger reinforcing effect, in the case of the PLA/PHA composite, is caused by the more extensive load-bearing capacity of the long PHA nanofibrils forming the network as well as the existence of larger interfacial areas between the nanofibers and the matrix. The main contribution to a significant increase in elastic and strength properties is associated with the formation of the PHA nanofibril network. This clearly demonstrates the dependences of true stress as a function of the Hencky strain after subtracting the data for neat PLA from respective stress–elongation curves for the nanocomposite (Appendix A). It is revealed that the nanofibril’s response is weak for small strains. However, exceeding the strain of 0.5, a strong strain hardening occurs which could be attributed to the straining of the network of PHA entangled nanofibers. 

The formation of PHA nanofibers may also be responsible for an essential increase in the ductility of the PLA matrix. These PHA nanofibers, along with PLA nanofibers spanning the craze surfaces, dissipate a large amount of energy until they break, providing a continuous crack nano-bridge toughening right before the crack front. Figure 6 and Figure 7 show the SEM images of crazes for the PLA/PHA blend and composite. For neat PLA, the crazes are thin, 1–2 μm, and do not thicken until failure [11]. Crazes in the PLA/PHA blend are almost the same size, 1–4 μm (Figure 6). The slightly larger craze size is due to the fact that the PHA nanoparticles may toughen the matrix and facilitate the PLA nanofibers’ formation during the development of crazes. However, for the PLA/PHA composite, crazes thicken with increasing strain up to a few tens of μm (Figure 7). 

It has been shown in our previous paper [11] that neat PLA demonstrates a quite brittle fracture with severe localization of strain before yielding. The addition of 3 wt.% PHA to PLA increases the number of crazes. Despite some crack nano-bridge toughening by PLA nanofibers, which makes it possible to transfer loads to the rest of the material and avoid overloading heavily loaded crazes, the deformation mechanism remains crazing. Nevertheless, a certain amount of crazes provides some load-bearing capacity and leads to the improvement of the ductility of PLA/PHA blends. For the PLA/PHA composite, a neck appears after yielding. In the initial stages of deformation (e ≤ 0.20), intensive crazing is observed, but then at higher values of deformation (e > 0.22), crazing is replaced by the plastic flow of the material (Figure 8).

It is known that neat PLA exhibits a strong strain softening, which stimulates strain localization and causes the build-up of local tri-axial stress. Since the local strain of neat PLA could not be delocalized, the local tri-axial stresses will induce void nucleation and crazes in the matrix leading to a brittle fracture behavior [27,28]. In the PLA/PHA blend, as well as the PLA/PHA composite, these tri-axial stresses could be released via intensive crazing i.e., the local strain in the PLA matrix would be delocalized. At a certain density of these crazes, the stress state in the PLA matrix could be converted from tri-axial to uni-axial. The new stress state is favorable for the initiation of shear bands leading to shear yielding of the matrix, as can be seen in Figure 9. Such a change in deformation mode from crazing to shear yielding was observed by us for the PLA/PBS system [11]. 

## 4. Conclusions

The concept of the in situ generation of polymer–polymer nanocomposites was applied to a PLA/PHA blend. Green biopolymer–biopolymer nanocomposites combining biodegradability, biocompatibility, good strength of PLA, and good toughness of PHA were obtained. It was shown that a high shear rate allows the formation of nano-sized spherical particles of PHA at the stage of mixing, and then simultaneous fibrillation and crystallization at the extrusion stage at a higher temperature, without the necessity for succeeding cooling below the temperature of their non-isothermal crystallization. Shearing of PHA with 850 s^−1^ increased the non-isothermal crystallization temperature by 50 °C and decreased the temperature range in which the transition from a molten state to a crystallized one occurs by 17 °C. Using the slit capillary increased the residence time in the extruder and facilitated the formation of a fibril network at a low concentration of PHA. The formed PHA fibers had a diameter of nearly 200 nm and acted not only as reinforcements for the polymeric matrix, but also as toughening elements by providing continuous crack nano-bridging right before the crack front, resulting in an increase of both strength and plasticity of PLA. In particular, the Young modulus, stress at break, and elongation at break exceeded the related parameters of neat PLA by 15, 30, and 400% (2.35 GPa, 56.0 MPa, and 28.1%), respectively.

## Figures and Tables

**Figure 1 polymers-11-01811-f001:**
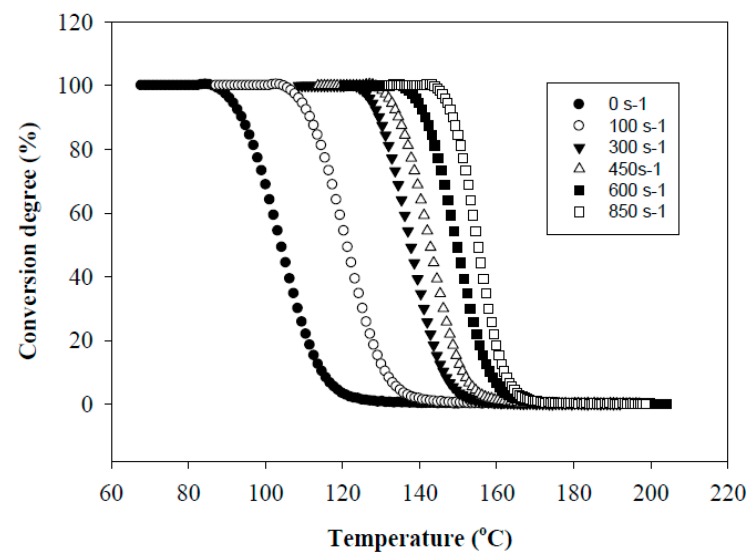
Dependencies of conversion degree of polyhydroxyalkanoate (PHA) during cooling with simultaneous shearing with various shearing rates. The conversion of melt into the crystalline phase was followed by recording the intensity of transmitted depolarized light.

**Figure 2 polymers-11-01811-f002:**
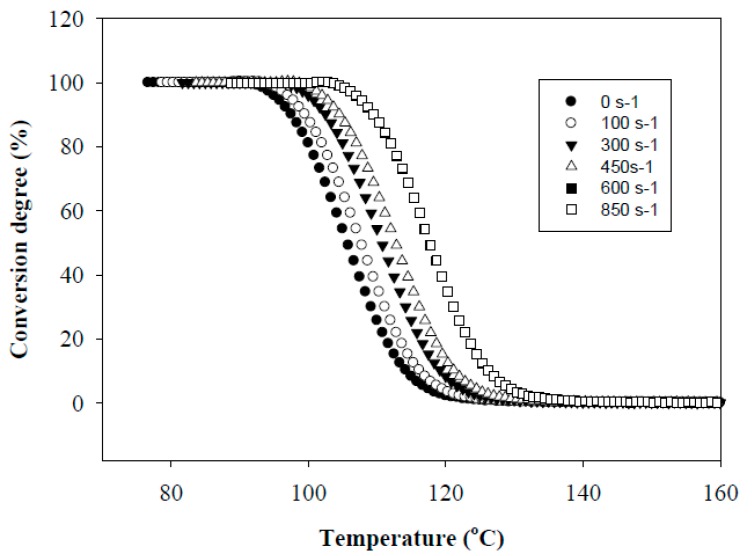
Dependencies of conversion degree of PHA during cooling with pre-crystallization shearing.

**Figure 3 polymers-11-01811-f003:**
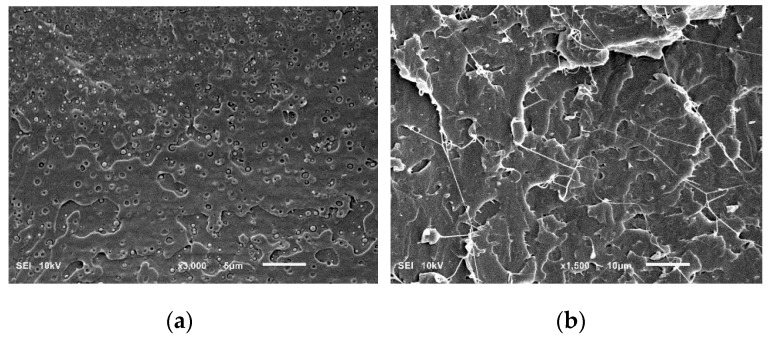
SEM images of cryofracture surfaces of polylactide (PLA)/PHA blends (**a**) and in situ generated composites (**b**).

**Figure 4 polymers-11-01811-f004:**
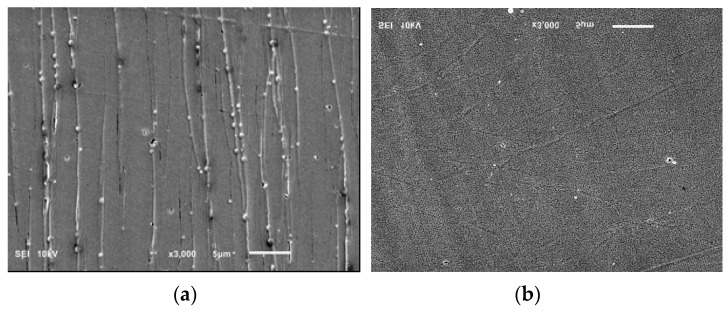
SEM images of ultramicrotomed surfaces of in situ generated PLA/PHA composites. (**a**) transverse, (**b**) longitudinal sections.

**Figure 5 polymers-11-01811-f005:**
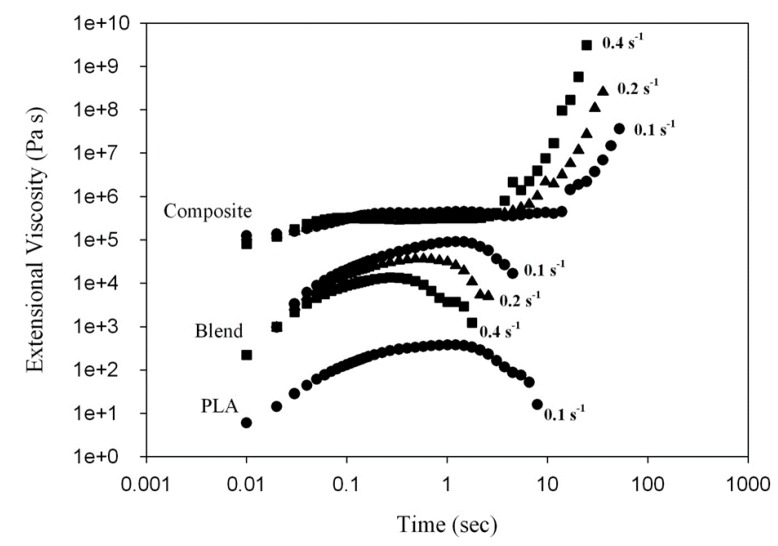
Time-dependence of the tensile stress growth coefficient, $\eta_{E}^{+}$ (t, έ), for molten PLA, the PLA/PHA blend and in situ generated composite.

**Figure 6 polymers-11-01811-f006:**
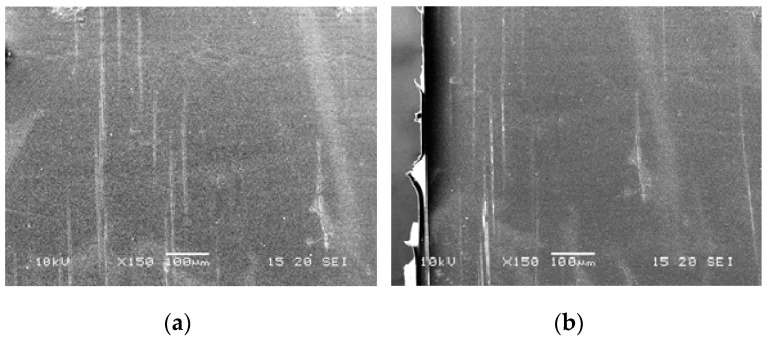
In situ observation of craze formation in the PLA/PHA blend originated by crack nano-bridge toughening via PLA fibers. Uniaxial deformation inside SEM microscope. The respective deformations are 0.049 (**a**) and 0.071 (**b**).

**Figure 7 polymers-11-01811-f007:**
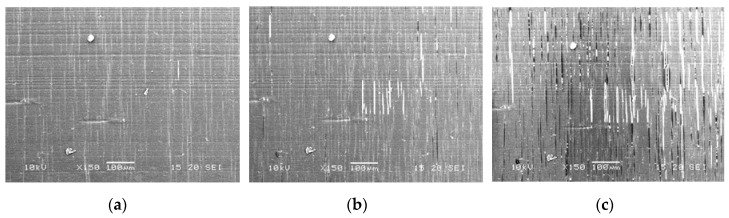
In situ observation of craze formation in the PLA/PHA composite originated by crack nano-bridge toughening via PHA fibers. Uniaxial deformation inside SEM microscope. The respective deformations are (**a**) 0.091, (**b**) 0.140, and (**c**) 0.182.

**Figure 8 polymers-11-01811-f008:**
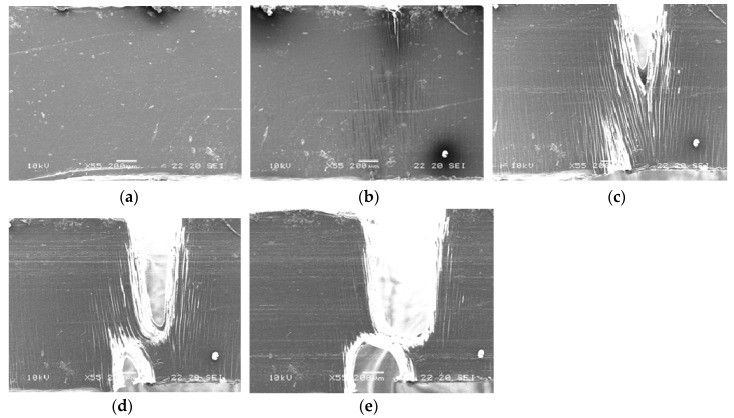
In situ SEM micrographs of the structure evolution of the in situ generated PLA/PHA composite subjected to a tensile test. The respective deformations are (**a**) 0.020, (**b**) 0.077, (**c**) 0.166, (**d**) 0.223, and (**e**) 0.247.

**Figure 9 polymers-11-01811-f009:**
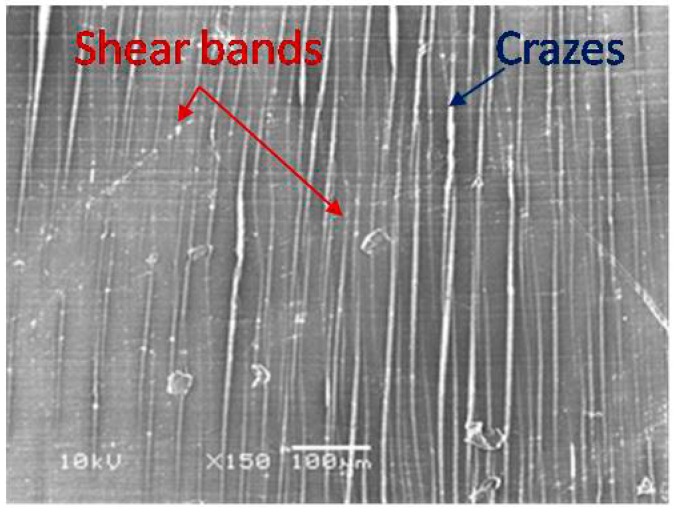
Simultaneous formation of crazes and shear bands.

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
