# Peer review of "Nanofibrillar Green Composites of Polylactide/Polyhydroxyalkanoate Produced in Situ Due to Shear Induced Crystallization"

_polymers, 2019, doi:10.3390/polym11111811_

Round 1

Reviewer 1 Report

“Nanofibrillar Green Composites of  Polylactide/Polyhydroxyalkanoate Produced in Situ  due to Shear Induced Crystallization”

This work is interesting in the area of polymer processing and can be considered for publication after major revision as follows:

There are few grammatical errors throughout the manuscript. These should be corrected. It would be interesting to verify micro or nanofibrils structures using some theoretical model. Mechanism of transformation of the dispersed PHA phase into nanofibrils has to be provided. This is needed for this manuscript. “In the case in-situ generated PLA/3wt.%PHA composite сhatacterized by fibrillar morphology Young modulus, stress at break and elongation at break exceeded the related parameters  of neat PLA by 15, 30, and 400% (2.35 GPa, 56.0 MPa, and 28.1%) respectively. This could be explained  by the presence of PHA nanoparticles in PLA matrix which restrict the mobility of PLA chains and  thus improve the stifness of the PLA matrix”. This can also be explained from improved inter-domain interaction between PHA nanoparticles and PLA. Lower domain size (nanometric range) lead to higher contract surface area between two phases of polymer blends leading to increase in stiffness/modulus. I recommend authors to see these articles for extended explanation of the above statements (Macromolecular Materials and Engineering, 2015, 300, 283-290 and Industrial & Engineering Chemistry Research, 2015, 54, 8137-8146).

Author Response

Comments:

This work is interesting in the area of polymer processing and can be considered for publication after major revision as follows:

1) There are few grammatical errors throughout the manuscript. These should be corrected.

Following the comment, the manuscript was proofread and checked using the Grammarly to edit any grammar, spelling, punctuation mistakes. The final version of manuscript, after applying the whole corrections, is attached subsequently.

2) It would be interesting to verify micro or nanofibrils structures using some theoretical model.

As a model we suggested to consider the dependences of true stress as a function of the Hencky strain after subtracting the data for neat PLA from respective stress-elongation curves for the nanocomposite (Supplementary Figure 4). It is revealed that the nanofibril’s response is weak for small strains. However, exceeding the strain of 0.5, a strong strain hardening occurs which is attributed to the straining of the network of PHA entangled nanofibers.

3) Mechanism of transformation of the dispersed PHA phase into nanofibrils has to be provided. This is needed for this manuscript.

A more detailed description of transformation of the dispersed PHA phase into nanofibrilsis added to the revised manuscript. Page 5, lines 170-180

4) "In the case in-situ generated PLA/3wt.%PHA composite сhatacterized by fibrillar morphology Young modulus, stress at break and elongation at break exceeded the related parameters  of neat PLA by 15, 30, and 400% (2.35 GPa, 56.0 MPa, and 28.1%) respectively. This could be explained by the presence of PHA nanoparticles in PLA matrix which restrict the mobility of PLA chains and thus improve the stifness of the PLA matrix". This can also be explained from improved inter-domain interaction between PHA nanoparticles and PLA. Lower domain size (nanometric range) leads to higher contract surface area between two 
phases of polymer blends leading to increase in stiffness/modulus. I recommend authors to see these articles for extended explanation of the above statements (Macromolecular Materials and Engineering, 2015, 300, 283-290 and Industrial & Engineering Chemistry Research, 2015, 54, 8137-8146).

The suggested explanation with relative reference was added to the manuscript to realize the effect of PHA nanofibers on the improved mechanical properties of composite. Page 7, lines 255-261

Reviewer 2 Report

The manuscript seems interesting and from my point of view could be published.

However, consulting a specialist in polymer rheology could help the publisher in making the right decision.

Author Response

Comments:
The manuscript seems interesting and from my point of view could be published.
However, consulting a specialist in polymer rheology could help the publisher in making the right decision.

Round 2

Reviewer 1 Report

Revised version can be considered for publication.